# JudgeRank: Leveraging Large Language Models for Reasoning-Intensive Reranking

## Abstract

Accurate document retrieval is crucial for the success of retrieval-augmented generation (RAG) applications, including open-domain question answering and code completion. While large language models (LLMs) have been employed as dense encoders or listwise rerankers in RAG systems, they often struggle with reasoning-intensive tasks because they lack nuanced analysis when judging document relevance. To address this limitation, we introduce JudgeRank,[1] a novel agentic reranker that emulates human cognitive processes when assessing document relevance. Our approach consists of three key steps: (1) *query analysis* to identify the core problem, (2) *document analysis* to extract a query-aware summary, and (3) *relevance judgment* to provide a concise assessment of document relevance. We evaluate JudgeRank on the reasoning-intensive BRIGHT benchmark, demonstrating substantial performance improvements over first-stage retrieval methods and outperforming other popular reranking approaches. In addition, JudgeRank performs on par with fine-tuned state-of-the-art rerankers on the popular BEIR benchmark, validating its zero-shot generalization capability. Through comprehensive ablation studies, we demonstrate that JudgeRank's performance generalizes well across LLMs of various sizes while ensembling them yields even more accurate reranking than individual models.

## 1 Introduction

Passage reranking is a critical component in modern information retrieval systems, designed to refine results obtained from efficient first-stage retrieval methods such as BM25 (Robertson et al., 1995; 2009). By narrowing down the pool of candidate documents, reranking substantially improves the quality of downstream tasks, such as retrieval-augmented generation or RAG (Lewis et al., 2020). Two primary approaches have emerged to address the reranking task. The first category comprises encoding-based approaches (Nogueira & Cho, 2019; Gao et al., 2021), which encode queries and documents into fixed-size embedding vectors. These methods use either cosine similarity as a score function or directly output a score from the model (Nogueira et al., 2020; Zhuang et al., 2023). While highly efficient, these approaches face several limitations. One major challenge is their inflexibility in defining relevance, making it difficult to accommodate diverse retrieval objectives (e.g., finding *supporting* vs. *refuting* evidence). Moreover, encoding-based models heavily rely on manual supervision signals due to the discrepancy between LLM pretraining and reranking objectives, limiting their ability to generalize to new domains or models (Nguyen, 2016; Izacard et al., 2022).

Most recently, utilizing Large Language Models (LLMs) for document reranking has led to promising progress in addressing some of these challenges, owing to their superior capabilities in language understanding, generation, interaction, and reasoning (Ouyang et al., 2022). These approaches utilize an LLM either as a pointwise judge (Ma et al., 2024) or a listwise reranker (Sun et al., 2023; Zhuang et al., 2024). While these approaches allow for flexible definition of document relevance and support zero-shot operation, they still require the model to make decisions without intermediate analyses. Consequently, they fall short in scenarios requiring complex reasoning (Su et al., 2024), hampering both performance and interpretability. Moreover, listwise rerankers face significant computational challenges due to context length constraints, often compromising on individual document length when processing multiple documents simultaneously.

---

[1]We plan to release our code upon acceptance.

Figure 1: A step-by-step illustration of how JUDGERANK arrives at the final judgment through query and document analyses. The query analysis identifies the core problem being asked, while the document analysis extracts relevant sentences from the document based on the query. This is a real example from the Biology task in the BRIGHT evaluation benchmark.

To bridge this gap, we propose JUDGERANK, a novel zero-shot pointwise reranker tailored for reasoning-intensive text retrieval tasks. Inspired by Chain-of-Thought (Wei et al., 2022) and LLM-as-a-Judge (Zheng et al., 2023) methods, JUDGERANK utilizes highly generalizable prompts to guide instruction-tuned LLMs through explicit reasoning steps before arriving at a final judgment.

Figure 1 illustrates a real example of how our model works on the Biology dataset in the BRIGHT (Benchmark for Reasoning-Intensive Generative Retrieval Tasks) benchmark (Su et al., 2024). Specifically, our reranker first prompts the LLM to identify the core problem in the query, allowing it to focus on the central question while filtering out irrelevant context. Next, the model produces an extractive summary for each of the candidate documents and explains how it addresses the query. Finally, the model makes a relevance judgment based on the previous analyses. This process closely mimics how humans approach questions: by first skimming the document, identifying relevant parts, and then carefully reading these parts to obtain an answer (Masson, 1983). This structured pipeline enables JUDGERANK to transcend surface-level lexical matching, leveraging deeper semantic understanding to improve reranking accuracy.

We evaluate JUDGERANK on the recently constructed BRIGHT benchmark, widely regarded as one of the most challenging retrieval evaluation datasets. Despite the poor performance of state-of-the-art text embedding models and rerankers on this benchmark, our method achieves significant improvements over all existing baselines and secures the top position on the BRIGHT benchmark leaderboard among 89 models, surpassing the previous best model by a significant margin (9 points). Our work is the first to show that a zero-shot pointwise reranker can outperform a well-trained listwise reranker by a significant margin, challenging the common belief that listwise rerankers in general perform better than pointwise rerankers (Déjean et al., 2024). Additionally, we demonstrate that JUDGERANK readily generalizes to other popular retrieval benchmarks such as BEIR and performs competitively with state-of-the-art rerankers. We also analyze the complementarity of models at different scales by investigating the alignment of their ranking decisions. We observe that models of different sizes demonstrate a surprisingly orthogonal behavior on their relevance judgments, leading to a simple ensembling strategy that allows us to combine multiple models flexibly and achieve considerable performance gains on the final ranking.

**Query Analysis Prompt**

You will be presented with a/an *{query name}*.

Your task consists of the following step:

1. Analyze the *{query name}*:
- Carefully read each sentence of the *{query name}*.
- Identify the core problem or question being asked.

Here is the *{query name}*:
*{query}*

(a). Query Analysis Prompt of JudgeRank

**Document Analysis Prompt**

You will be presented with a/an *{query name}*, an analysis of the query, and a/an *{doc name}*.

Your task consists of the following steps:
1. Analyze the *{doc name}*:
- Thoroughly examine each sentence of the *{doc name}*.
- List all sentences from the *{doc name}* that *{definition of relevance}* the *{query name}*.
- Briefly explain how each sentence listed *{definition of relevance}* the *{query name}*.

2. Assess overall relevance:
- If the *{doc name}*, particularly the relevant sentences (if applicable), *{definition of relevance}* the *{query name}*, briefly explain why.
- Otherwise, briefly explain why not.

Here is the *{query name}*:
*{query}*

Here is the analysis of the *{query name}*:
*{query analysis}*

Here is the *{doc name}*:
*{doc}*

(b). Document Analysis Prompt of JudgeRank

**Judgment Prompt**

You will be presented with a/an *{query name}*, an analysis of the *{query name}*, a/an *{doc name}*, and an analysis of the *{doc name}*.

Your task is to assess if the *{doc name}* *{definition of relevance}* the *{query name}* in one word:
- Yes: If the *{doc name}* *{definition of relevance}* the *{query name}*.
- No: Otherwise.

Important: Respond using only one of the following two words without quotation marks: Yes or No.

Here is the *{query name}*:
*{query}*

Here is the analysis of the *{query name}*:
*{query analysis}*

Here is the *{doc name}*:
*{doc}*

Here is the analysis of the *{doc name}*:
*{doc analysis}*

(c). Judgment Prompt of JudgeRank

Figure 2: (a) Prompt to analyze query, where {query name} (e.g., "Biology post") and {query} are placeholders for the query type and content. (b) Prompt for analyzing a document, where {doc name} (e.g., "document") and {doc} are placeholders for the document type and content. (c) Prompt for making the final one-word relevance judgment.

## 2 METHOD

### 2.1 AGENTIC STEPS

Mimicking human cognitive process, JUDGERANK consists of three main steps: *Query Analysis*, *Document Analysis*, and *Relevance Judgment*. The prompt templates are illustrated in Figure 2.

**Query analysis** The query analysis prompt (Figure 2 (a).) directs the LLM to analyze the query by identifying the core problem being asked. Note that this prompt only depends on the query so that we can generate the query analyses separately and store them. Since the number of queries $n_q$ is usually much smaller than that of documents $n_d$ ($n_q \ll n_d$), we can afford to use a more expensive LLM (e.g., GPT-4) to handle this important step, and leave the other steps to relatively smaller LMs.

**Document analysis** The document analysis prompt (Figure 2 (b).) asks the LLM to output an extractive summary of the document that helps answer the query, and assess the overall relevance of the document based on the summary.

**Relevance judgment** The judgment prompt (Figure 2 (c).) asks the model to make a one-word judgment, either "Yes" or "No". We isolate this step to make it easier to ensemble with different judgment prompts or models.

### 2.2 GENERALIZABILITY OF THE PROMPTS

All three steps consume natural language as input and generate a response, making it more flexible to transfer and stack across different LLMs. The templates also show that the prompts are highly generalizable: to adapt them to a new reranking task, one only needs to replace the query name, the document name, and the relation between them.[2] Leveraging LLMs in a zero-shot setting allows

---

[2]Intuitively, there is a trade-off between prompt generalizability and reranking performance. To further push reranking accuracy, one always has the option to further adapt the prompt templates to the target tasks.

us to flexibly define the relation between the query and the document. This flexibility is important because the user may define either "document that supports a query" or "document that refutes a query" as the relation, which are opposites of each other. Encoding-based models usually cannot achieve such behavior zero-shot because most of them use cosine similarity or metrics alike to represent "relevance". One way encoding-based models could achieve such flexibility is through extensive fine-tuning. However, this requires additional training data and introduces new model parameters, potentially causing an unintended distribution shift. Similarly, models like RANKZEPHYR also finetunes the LLM to output document ids, thus suffering from a distribution shift as well.

## 2.3 METHODOLOGY OF SCORING DOCUMENTS

**Binary version**    The binary version creates a binary partition between accepted (when the model outputs a "Yes") and rejected (when the model outputs a "No") documents, maintaining the first-stage retrieval ranking within each category. More specifically, let $D = \{d_1, d_2, \ldots, d_k\}$ be an ordered list of top-$k$ documents ranked by the first-stage retrieval model. Let $D_y$ and $D_n$ be a partition of $D$, such that

$$D_y \cup D_n = D \tag{1}$$
$$D_y \cap D_n = \emptyset \tag{2}$$

, where $D_y$ is the set of documents that the reranker judged as relevant and $D_n$ is the set of documents that the reranker judged as non-relevant. Let $R$ be the reranking function which maps each document $d$ to its rank (lower rank means "more relevant"), then

$$\forall d \in D_y \text{ and } d' \in D_n, \ R(d) < R(d') \tag{3}$$

, and for the relative ranking within each partition,

$$\forall d_i, d_j \in D_y, \ R(d_i) < R(d_j) \iff R_0(d_i) < R_0(d_j) \tag{4}$$

, where $R(d)$ is the final rank of document $d$ and $R_0(d)$ is the rank of $d$ from the first-stage retrieval. The same applies to $D_n$.

While straightforward and well-performing, this approach is sensitive to prompt wording and relies heavily on first-stage retrieval performance. That said, for proprietary LLMs that do not allow access to the probabilities of the generated tokens, this approach is also the only option.

**Continuous version**    The continuous version addresses the limitations of the binary version by using the probability of the "Yes" judgment $p_y$ and the probability of the "No" judgment $p_n$ to construct a complete ranking. The probabilities are computed by first obtaining their log probabilities,[3] and then take their exponential. This is similar to the relevance generation approach proposed in Liang et al. (2023). The scoring function $S$ is defined by normalizing the probabilities between $p_y$ and $p_n$ as follows:

$$S(d) = \frac{p_y}{p_y + p_n} \tag{5}$$

This normalization step is necessary because the combined probability mass allocated to $p_y$ and $p_n$ is not always a fixed value across different documents. Without normalization, the $p_y$ values for different documents would not be directly comparable.

The final ranking $D_R$ is then defined as $D_R = \{d_1, d_2, \ldots, d_k\}$ such that

---

[3]One can easily obtain the log probabilities for open-source LLMs and proprietary models that provide this functionality, such as OpenAI.

$$\forall i, j \in \{1, 2, \ldots, k\}, \ i < j \iff S(d_i) > S(d_j) \tag{6}$$

This continuous version provides a more fine-grained ranking compared to the binary partition, as it utilizes the full range of probabilities output by the LLM.

**Hybrid version**   Additionally, we explore a variant of the continuous version where the final score is computed by taking a weighted sum of the probability score $S_{\text{prob}}$ and the BM25 score $S_{\text{BM25}}$. More specifically, the final score is computed by

$$S = \alpha S_{\text{prob}} + S_{\text{BM25}} \tag{7}$$

, where $S_{\text{BM25}}$ is the score provided by BM25 in the first-stage retrieval, and $\alpha$ is the relative weight of the probability score. We set $\alpha = 100$ in this work to bring $S_{\text{prob}}$ to the same scale as $S_{\text{BM25}}$. This version leverages model ensembling to consider both reasoning and surface-level matching, thus marrying the benefits of both approaches. Unless otherwise specified, we use this setting to compute the final score throughout the paper. In the ablation studies, we compare these three settings and show their relative performances.

## 3 EXPERIMENTAL SETUP

### 3.1 DATASETS

**BRIGHT**   We use the BRIGHT benchmark (Su et al., 2024) to assess the performance of our reranker. BRIGHT is specifically designed to evaluate text retrieval systems on complex, reasoning-intensive queries that go beyond simple keyword matching. The benchmark comprises 1,398 real-world queries spanning diverse domains, including economics, psychology, robotics, math, and software engineering. These queries are carefully curated to represent challenging scenarios that require deep understanding and reasoning to identify relevant documents. We use this dataset to evaluate our approach because unlike traditional benchmarks that focus on simple information-seeking tasks, BRIGHT queries require complex reasoning to determine document relevance, making it an excellent tool for evaluating advanced retrieval systems in realistic scenarios. The benchmark has also been validated to be robust against potential data leakage, maintaining its effectiveness even when benchmark documents have been included in model training data.

Because of its challenging nature, state-of-the-art retrieval models have shown significantly lower performance on BRIGHT compared to other benchmarks (Su et al., 2024). For example, the leading model on the MTEB leaderboard (Muennighoff et al., 2022) achieves an nDCG@10 of only 18.0 on BRIGHT, compared to 59.0 on other benchmarks. The GPT-4 listwise reranker also only improves around 2 points on nDCG@10 on top of the BM25 first-stage retrieval, while Gemini (Team et al., 2023) features less improvement than that. The cross-encoder reranker MiniLM (Wang et al., 2020) even significantly underperforms the BM25 baseline.

**BEIR**   To test the generalizability of our approach, we evaluate on the BEIR benchmark (Thakur et al., 2021), a robust and heterogeneous evaluation benchmark for information retrieval. We evaluate on all tasks that are publicly available (Kamalloo et al., 2023). For all datasets we use the the test set, except for MSMARCO where we follow BEIR convention to evaluate on the dev set. For *cqadupstack* we follow BEIR convention and evaluate on all sub-datasets and compute their average. Because BEIR is a large benchmark, and the largest dataset has more than 13K queries, we only generate query analysis to evaluate on this dataset. This almost adds no overhead to the generation because the query analysis generation does not depend on the document. Since our model only generates a single token "Yes" or "No", its latency is almost the same as encoding both the query and the document with an encoding-based retrieval model, making it a highly efficient alternative.

### 3.2 FIRST-STAGE RETRIEVAL

For both benchmarks we evaluate on, we follow common settings from previous work to rerank the top-100 documents from the first-stage retrieval and use nDCG@10 score as the evaluation metric.

This metric assesses the quality of the retrieved documents, taking into account both their relevance and ranking position.

**BRIGHT** The original BRIGHT paper explores using LLMs to generate Chain-of-Thought (Wei et al., 2022) reasoning steps as queries (Su et al., 2024), resulting in up to 12.2 point improvements on average.[4] We thus build on top of this best first-stage retrieval model on the leaderboard, which achieves an nDCG@10 score of 26.47 with BM25 and reasoning chains generated by GPT-4-0125-preview.

**BEIR** We follow the original BEIR paper (Thakur et al., 2021) and use ElasticSearch BM25 (Elasticsearch, 2018) as the first-stage retriever.

### 3.3 BASE MODEL

Our main model builds on top of Llama-3.1-70B-instruct (Dubey et al., 2024). We choose the Llama-3.1 model family because its RoPE scaling Liu et al. (2024b) allows longer context length up to 128K, which is essential in handling long documents. To speed up experiments, we evaluate on a quantized version of this model, namely Llama-3.1-70B-instruct-awq-int4. We also perform ablation studies where we evaluate on Llama-3.1-8B and Llama-3.1-405B-instruct-awq-int4.[5]

### 3.4 BASELINE RERANKERS

We reproduce RANKLLAMA (Ma et al., 2024) and RANKZEPHYR (Pradeep et al., 2023), two state-of-the-art rerankers as evaluated by the BEIR benchmark. RankLlama is a pointwise reranker that directly outputs a score. This model is trained on the MS MARCO passage ranking dataset (Bajaj et al., 2016). RANKZEPHYR is a listwise reranker that takes a query and a list of documents together as input and outputs a ranking. This model uses the queries sourced by Sun et al. (2023) from the MS MARCO dataset to distill GPT-3.5 and GPT-4 in sequence. We use the RERANKERS library (Clavié, 2024), a lightweight unified API that allows users to run diverse reranking models out-of-the-box.[6]

### 3.5 EFFICIENCY AND OPTIMIZATION

To make the encoding and generation more efficient, we use vLLM (Kwon et al., 2023), which leverages paged attention to improve throughput. Importantly, when designing the prompts used in our approach, we append the query and the document at the very end of each prompt to make the best use of Automatic Prefix Caching (Gim et al., 2024), which temporarily stores the KV cache of existing inputs so that a new input can directly reuse the KV cache if it shares the same prefix with one of the existing ones. This design greatly improves the efficiency of our experiments, and it is also the main motivation for us to choose decoupled analyses over Chain-of-Thought style prompts which ask the LLM to perform all the analyses and make a judgment in one take.

## 4 RESULTS AND DISCUSSION

### 4.1 MAIN RESULTS

**BRIGHT** As shown in Table 1, JUDGERANK achieves state-of-the-art results on the BRIGHT evaluation benchmark as measured by nDCG@10. Our best performing model improves upon the no-rerank baseline by more than 9 points, while RANKLLAMA underperforms the baseline and RANKZEPHYR stays barely above the baseline. The smaller Llama-3.1-8B-instruct also outperforms the baseline by more than 3 points, showing the generalizability of our approach across different model sizes. Interestingly, increasing model size from 70B to 405B does not bring a significant gain on nDCG@10. This is understandable because according to the benchmark performance

---

[4]See Table 38 of the original BRIGHT.

[5]For the 8B model we do not use the quantized version because it can already fit on a single A100 GPU, while the other two bigger models require quantization to save computational cost.

[6]https://github.com/AnswerDotAI/rerankers

| | BM25 | RankLlama | RankZephyr | JudgeRank | | | |
| --- | --- | --- | --- | --- | --- | --- | --- |
| | | | | 8B | 70B | 405B | Ensemble |
| Biology | 53.63 | 11.05 | 44.37 | 54.44 | 57.59 | 60.33 | **60.70** |
| Earth Science | 53.65 | 11.83 | 35.24 | 54.62 | 58.36 | 55.11 | **58.72** |
| Economics | 24.28 | 9.56 | 24.44 | 28.04 | 33.01 | 32.15 | **35.39** |
| Psychology | 38.59 | 11.38 | 36.92 | 42.16 | 46.59 | 45.42 | **47.57** |
| Robotics | 18.77 | 8.53 | 18.80 | 24.09 | **28.30** | 27.64 | 28.16 |
| Stack Overflow | 22.74 | 11.40 | 19.62 | 27.18 | 27.47 | 28.30 | **29.74** |
| Sustainable Living | 25.90 | 11.75 | 29.38 | 30.69 | 39.55 | 38.54 | **41.88** |
| Leetcode | 19.27 | 20.32 | **24.58** | 17.49 | 20.06 | 22.47 | 20.23 |
| Pony | 17.73 | 18.88 | **48.96** | 22.85 | 30.82 | 31.54 | 32.74 |
| Aops | 3.92 | 3.55 | 6.98 | 6.15 | 8.24 | 7.74 | **8.57** |
| TheoremQA-Questions | 18.90 | 11.82 | 22.34 | 24.09 | 23.83 | **26.93** | 25.86 |
| TheoremQA-Theorems | 20.22 | 5.63 | 7.78 | 29.89 | 34.55 | 36.16 | **36.20** |
| Average | 26.47 | 11.31 | 27.25 | 30.14 | 34.03 | 34.36 | **35.48** |

Table 1: JUDGERANK nDCG@10 results on the BRIGHT evaluation benchmark. Best results on each dataset and the entire benchmark are boldfaced. "Ensemble" stands for model ensembling of JUDGERANK-8B, 70B, and 405B.

presented by Meta[7], there are multiple tasks where 405B is not significantly stronger than 70B, such as COMMONSENSEQA, TRIVIAQA-WIKI, and BOOLQ. We hypothesize that the BRIGHT benchmark also falls in this plateauing category. We thus select the 70B version as our main model to balance between efficiency and performance. Our results also show that in zero-shot settings, our pointwise reranker is better than listwise approaches for reasoning-intensive tasks. For example, the original BRIGHT paper shows that GPT-4 with zero-shot listwise reranking improves on top of vanilla BM25 baseline by an average of 2.7 points on nDCG@10, a much smaller improvement compared to JUDGERANK despite using a much stronger LLM. To examine if our prompts are sensitive to wordings, we additionally conduct an experiment where we use the quantized version of Llama-3.1-70B-instruct to paraphrase all three prompt templates and conduct the reranking experiment with Llama-3.1-8B. We find that the model achieves an nDCG@10 of 30.36, which is very close to 30.14 as reported in Table 1, showing that our approach is not sensitive to prompt variations.

**BEIR** As shown in Table 2, our model delivers competitive results on the BEIR evaluation benchmark despite the fact that RANKLLAMA and RANKZEPHYR are heavily fine-tuned on in-domain data including MS MARCO, which is part of the BEIR benchmark. Note that for the less reasoning-intensive BEIR benchmark, we do not include the document analysis step, which is the only step that cannot be KV-cached in advance. This setting shows that one can flexibly adjust the complexity of our approach based on the complexity of the task. Overall, there is a much smaller performance gap on BEIR between JUDGERANK and the baselines than on BRIGHT, indicating that our approach is more geared toward reasoning-intensive tasks.

## 4.2 DISCUSSION

We pose several research questions to illustrate whether and how our approach works.

**How complementary are LLMs of different scales?** In Table 1, we observe that JudgeRank-70B and JudgeRank-405B performs on par with each other. However, nDCG@10 alone does not reveal the whole picture. One natural question to ask is: do these two models make similar judgments or are complementary to each other? To answer this question, we obtain statistics on the percentage of both models agreeing and disagreeing each other and show them on the left of Figure 3. From the tables we can see that for all three combinations of the models, the majority case is always that both models rejects the documents. This is understandable because only a few out of the top-100 documents are supposed to be relevant. The interesting pattern emerges when we inspect the other three cases: each pair of the models spends more time disagreeing with each other than both outputting "Yes". For the pairs 8B vs 70B and 8B vs 405B, there is a higher difference because the

---

[7]https://huggingface.co/meta-llama/Llama-3.1-405B

|                | BM25      | RankZephyr | RankLlama | JudgeRank |
|----------------|-----------|------------|-----------|-----------|
| webis-touche2020 | **34.71** | 33.34      | 32.97     | 27.94     |
| trec-covid     | 68.80     | **85.77**  | 81.08     | 83.70     |
| scifact        | 69.06     | **74.68**  | 75.57     | 73.24     |
| nfcorpus       | 34.28     | **38.79**  | 35.92     | 37.73     |
| dbpedia-entity | 32.02     | 44.19      | 43.72     | **44.30** |
| fiqa           | 25.36     | 40.40      | **42.70** | 40.35     |
| scidocs        | 16.47     | **19.70**  | 19.26     | 19.47     |
| arguana        | 47.16     | 47.07      | 32.08     | **62.77** |
| nq             | 32.61     | **60.10**  | 59.19     | 56.87     |
| climate-fever  | 18.61     | **24.26**  | 17.75     | 19.00     |
| fever          | 64.94     | **79.52**  | 78.55     | 69.97     |
| msmarco        | 22.75     | 37.89      | **41.40** | 34.01     |
| hotpotqa       | 60.22     | **71.16**  | 70.65     | 68.30     |
| quora          | 80.77     | 79.60      | 82.73     | **84.25** |
| cqadupstack    | 32.53     | 42.68      | 42.43     | **43.98** |
| Average        | 42.69     | **51.94**  | 50.40     | 51.06     |

Table 2: JUDGERANK (70B) nDCG@10 results on the BEIR evaluation benchmark. Best results on each dataset and the entire benchmark are boldfaced.

capabilities of the two models differ more. In contrast, for 70B vs 405B there is less disagreement. From these observations, we indeed see that each two models may be complementary to each other.

Motivated by this observation, we take model ensembling one step further. Recall that in Section 2.3, we ensemble the BM25 score with each of the scores output by the Llama models. Here we first take the average score output by all the Llama models, and then perform the weighted sum with the BM25 score. More specifically, let $S_{8B}$, $S_{70B}$, and $S_{405B}$ be the score assigned by each model, respectively, the ensemble score of the three models is computed as $\alpha(S_{8B} + S_{70B} + S_{405B})/3 + S_{BM25}$, where again $\alpha = 100$ and $S_{BM25}$ is the score given by the BM25 model. The same equation generalizes analogously to two-model ensembles.

We present all ensembling results on the right of Figure 3. We can see that each ensembling performance is better than its individual model performances, with the strongest performance observed when ensembling all three models. This result shows that a salient performance boost can be achieved by ensembling two of the strongest models (70B + 405B), while even the model with lower performance (i.e., 8B) could contribute positively in model ensembling. Intuitively, such ensembling is equivalent to a verification or a majority voting step. The final score is the highest when both models say "Yes", the score is medium when one of the two says "No", and the lowest score is observed when both say "No".[8]

**How does the choice of reranking score impact the final performance?** Recall that to compute the final score, we take a weighted sum of the BM25 score and the probability score from the judgment step. To justify this choice, we compare it with two other settings: the first is binary judgment, and the second only uses the normalized probability to rerank documents (introduced in Section 3). The left part of Figure 4 shows that binary judgment performs the worst among the three settings while using only probability achieves somewhere in between. This is understandable because binary judgments are sensitive to wordings. Imagine that if we change the relation from "substantially helps answer" to "helps answer" or "at least partially helps answer," the number of "Yes" that the model outputs will keep increasing, thus also increasing the number of false positives. However, the other two settings are not sensitive to such changes.

**How useful are the query and document analysis steps?** To show the effectiveness of the analysis steps, we perform an ablation study on BRIGHT. We remove the two analysis steps and keep

---

[8]Model ensembling boosts performance at the price of higher latency. In practice, practitioners can decide on the most suitable configuration based on how sensitive the downstream tasks are to retrieval performance.

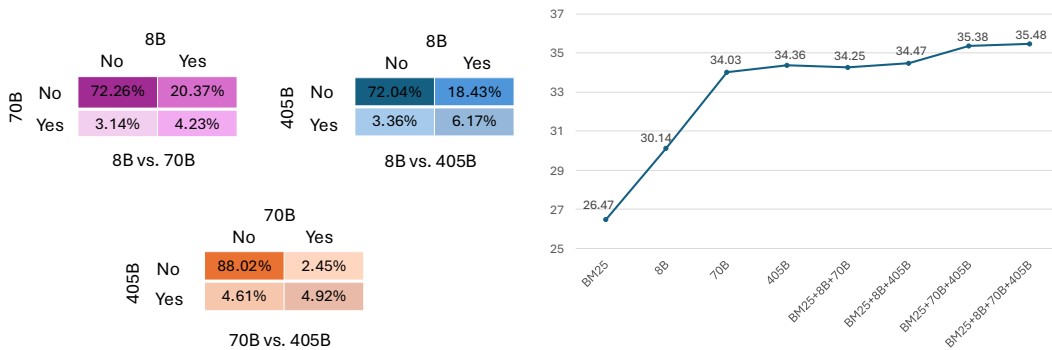

Figure 3: **On the left**: judgment alignment studies for models of three sizes: 8B, 70B, and 405B. Percentages are shown for each quadrant. **On the right**: nDCG@10 of each individual model and model ensembling on the BRIGHT evaluation benchmark.

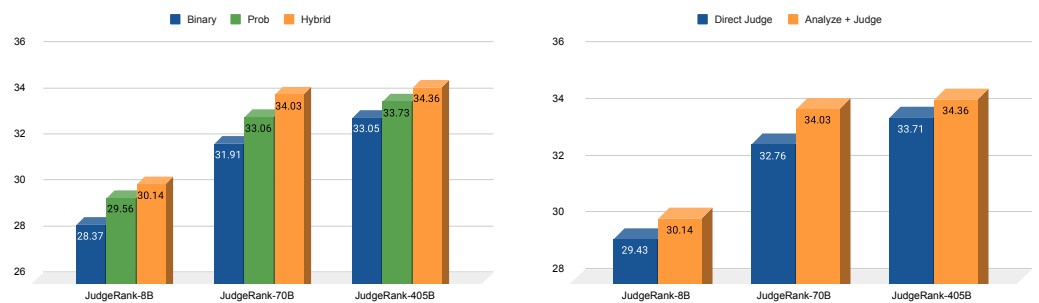

Figure 4: Ablation studies of JudgeRank. **On the left**: Comparison of three scoring settings on the BRIGHT evaluation benchmark. *Binary* stands for binary judgment, *Prob* stands for probability, and *Hybrid* stands for a weighted sum of BM25 and probability scores. **On the right**: Comparison of direct judge and judge with query and document analyses on the BRIGHT evaluation benchmark.

the judgment step untouched.[9] The right of Figure 4 shows that judging with query and document analyses performs consistently better than the direct judgment approach.[10]

**Qualitative examples**  Figure 5 demonstrates how JUDGERANK enhances document relevance identification using real examples from the BRIGHT dataset. In the left example, we observe a document initially ranked high by the first-stage retriever is correctly identified as irrelevant. Despite surface-level similarities, JUDGERANK fails to extract any sentences that help answer the query, revealing that the document is merely an advertisement coincidentally sharing common terminology with the query. The right example presents a contrasting scenario, where a document initially ranked low by the first-stage retriever is accurately identified as highly relevant. In this instance, the document analysis prompt enables the LLM to pinpoint key sentences that elucidate the underlying mechanism of funnel web spider venom's lethality, precisely addressing the query. These extracted sentences further inform the LLM to make the final positive judgment, demonstrating JudgeRank's ability to uncover deeply relevant content that might be overlooked by traditional retrieval methods.

## 5  RELATED WORK

The field of reranking models can be understood through two primary dimensions.

---

[9]This setting is highly efficient because its latency is practically identical to encoding-based models of the same size.

[10]The analysis steps involve generating tokens, which leads to higher latency. In practice, practitioners can decide on the most appropriate setup based on how sensitive the downstream tasks are to retrieval performance.

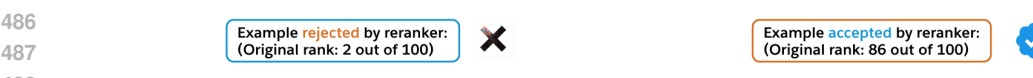

Figure 5: Illustration of how agentic generations of JUDGERANK help identifying the relevant documents. **On the left**, the document is ranked high by the first-stage retriever but judged as negative by the reranker. **On the right**, the document is ranked low by the first-stage retriever but judged as positive by the reranker because the document analysis prompt helps the LLM to locate the relevant sentences that answer the query.

The first dimension distinguishes between encoding-based and LLM-based approaches. Encoding-based models (Nogueira & Cho, 2019; Gao et al., 2021), typically require extensive training to adapt to the specific objectives of reranking tasks. In contrast, LLM-based models (Sun et al., 2023; Tang et al., 2024; Qin et al., 2024; Zhuang et al., 2024), demonstrate impressive zero-shot capabilities, allowing them to perform effectively without task-specific fine-tuning. Some work has also explored pretraining, fine-tuning and distillation techniques for LLM-based rerankers to further enhance their performance (Zhang et al., 2023; Ma et al., 2024; Yu et al., 2024).

The second dimension differentiates between pointwise and listwise reranking. Pointwise rerankers (Ma et al., 2024; Guo et al., 2024) evaluate the relevance of individual query-document pairs in isolation, producing a score for each pair without direct comparison between documents. In contrast, listwise rerankers (Sun et al., 2023; Pradeep et al., 2023; Ma et al., 2023; Yoon et al., 2024; Liu et al., 2024a; Tang et al., 2024) consider the entire set of documents for a given query, generating an ordered list as output. To address the challenges posed by input length limitations, researchers have developed innovative techniques such as sliding window approaches (Sun et al., 2023), which allow for the ranking of smaller subsets of documents before aggregating them into a comprehensive ranking. This framework encompasses pairwise (Pradeep et al., 2021; Qin et al., 2024) and setwise (Zhuang et al., 2024) rerankers as specific instances of the broader listwise category.

Our contribution is a novel LLM-based, pointwise reranker that leverages the capabilities of LLMs and incorporates explicit reasoning steps in the relevance judgment process. This approach sets our work apart from previous efforts by enhancing the model's ability to handle complex, reasoning-intensive reranking tasks while simultaneously improving the interpretability of its decisions.

## 6 CONCLUSION

In this work, we target document retrieval tasks that require intensive context-based reasoning. Through experiments and ablation studies, we show that our reranker outperforms previous state-of-the-art reranking models, while remaining flexible and efficient. In section 4.2 we have shown the significant benefit of model ensembling on document reranking. Yet we do not have to stop there. Other than ensembling LLMs of different model families, or even ensembling with encoder-only embedding models, there are at least two categories of ensembling that we envision. First, sampling ensembling: for each generation, we sample several generations, each of which could lead to a different judgment. This kind of ensembling is similar to the self-consistency approach (Tang et al., 2024). Second, prompt ensembling: we could leverage paraphrases of the same prompt to perform ensembling. We leave the exploration as future work because such approaches could be generalized to many prompting-related tasks, and thus better be addressed in separate works.

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
