# OpenReview forum: "JudgeRank: Leveraging Large Language Models for Reasoning-Intensive Reranking"
_ICLR.cc/2025/Conference — Submitted to ICLR 2025_

### Official Review · Reviewer_27LA · 2024-10-22

**Soundness:** 2
**Presentation:** 3
**Contribution:** 2
**Rating:** 5
**Confidence:** 4

**Summary:**

This paper presents JudgeRank, a LLM-based re-ranking framework for reasoning-extensive retrieval. It consists of three steps (components): query analysis, document analysis, and final judgement making. In query analysis, an LLM is prompted to generate the core problem or question of the query/question. In document analysis, an LLM is prompted to analyze every top-k retrieved document with the analysis of the query, and the definition of relevance. In final judgement making, an LLM is prompted to generate a Yes/No judgement given the query and document analysis, and the definition of relevance. Experiments show that JudgeRank works well in both reasoning-extensive and ordinary retrieval tasks.

**Strengths:**

The proposed method is simple and effective, and generalizes well on a series of retrieval tasks.

**Weaknesses:**

- The authors argue that “In contrast, the original BRIGHT paper shows that GPT-4 with listwise reranking improves on top of vanilla BM25 baseline by an average of 2.7 points on nDCG@10, a much smaller improvement than our approach despite using a much stronger LLM”. However, the experiments from the BRIGHT paper is conducted on vanilla BM25, without the use of CoT reasoning on the query, whereas JudgeRank in this paper reranks top documents from BM25 with query-side CoT reasoning. So the comparison here is not fair - “GPT-4 with listwise reranking” should be run in the same setting.
- The effectiveness of this three-step framework is not clear. What is the result of only removing the query or document analysis step? Do both analysis steps help? And, what about putting the document analysis and judgment into a single prompt, simplifying the process into two steps? An in-depth analysis of the design of JudgeRank is needed.
- See questions.

**Questions:**

- How do authors obtain the probability of the “Yes” judgment? This needs to be explained.
- The efficiency of JudgeRank is not examined in this paper. Including an analysis of its running time and a comparison of RankGPT and other LLM-based ranking methods in a fair setting would be better.
- What is “{definition of relevance}” set to in each task? How does it affect the re-ranking result?

---

> ### Author Response · Authors · 2024-11-23
>
> We thank Reviewer 27LA for the feedback.
>
> * Experiments from the BRIGHT paper is conducted on vanilla BM25, without the use of CoT reasoning on the query, whereas JudgeRank in this paper reranks top documents from BM25 with query-side CoT reasoning
>
> We respectfully disagree. As described in Section 3.2, we compare with and build on top of BM25 with CoT reasoning. There is no vanilla BM25 involved. The 26.47 on nDCG@10 for the BM25 + CoT baseline is reproduced from the released code from the BRIGHT authors (https://github.com/xlang-ai/BRIGHT) and it is also reported in the original BRIGHT paper (https://arxiv.org/pdf/2407.12883, Table 38, first line). We have modified Section 3.2 in the revision to point to Table 38 in the original BRIGHT paper.
>
> As this is a major misunderstanding of the baseline we compare with, we sincerely hope that the reviewer could consider raising our score after seeing this clarification. Happy to answer any follow-up questions to further clarify, if needed.
>
> * Latency of JudgeRank
>
> Each query corresponds to 100 documents to rerank. Our full version which includes query and doc analyses runs 127.57 seconds/query. The Direct Judge version runs 4.49 seconds/query. As a comparison, RankZephyr runs 49.6 seconds/query.
>
> * What is the result of removing the query or document analysis step?
>
> The ablation study for this is presented in right subfigure of Figure 4 and discussed in the last research question of Section 4.2.
>
> * How do authors obtain the probability of the “Yes” judgment? This needs to be explained.
>
> For vLLM we use the logprobs parameter (https://docs.vllm.ai/en/latest/dev/sampling_params.html) to obtain the log probability of the “Yes” token, and take the exponential of it to get the final probability. We have included this in Section 2.3 in the revision.
>
> * What is “{definition of relevance}” set to in each task?
>
> We will include the value of each placeholder in the Appendix section in the revision.

---

> > ### Comment · Reviewer_27LA · 2024-12-01
> >
> > Thank you for your response. I apologize for my previous misunderstandings. Considering that part of my concerns have not been addressed (the result of removing the query or document analysis step separately and putting the document analysis and judgment into a single prompt, and the impact of “{definition of relevance}” on the re-ranking performance), I decide to raise my score from 3 to 5.

---

> ### Author Response · Authors · 2024-12-03
>
> We thank Reviewer 27LA for updating the score! We are glad to have clarified some of the questions.
>
> To address the remaining concern, we have conduct two new experiments on BRIGHT with Llama-3.1-8B:
> (1) If we only apply query analysis, the model would obtain an nDCG@10 of 29.65, while if we only apply document analysis, the model would achieve 30.01. In this sense, document analysis is slightly more important than query analysis, at least for this setting.
>
> (2) When putting the query and document analysis in one single prompt, the model achieves 30.08, as compared to 30.14 where there are 3 separate prompts. Based on the small gap, it is likely worth it to modularize the prompts to leverage the prefix caching advantage to reduce latency.
>
> As December 2 (today) marks the final day for reviewers to communicate any messages to the authors. We sincerely hope that we have adequately addressed all your questions and concerns. If you feel that your feedback has been satisfactorily met, we would appreciate it if you could consider further updating your score. Thank you for your time and contributions!
>
> Best Regards,
>
> JudgeRank Authors

---

### Official Review · Reviewer_WdBJ · 2024-11-03

**Soundness:** 2
**Presentation:** 3
**Contribution:** 3
**Rating:** 5
**Confidence:** 4

**Summary:**

The paper introduces an LLM-based document ranking pipeline given a query. The authors use 3 main stages, query analysis, document analysis and judgment. They evaluate their pipeline on 2 main benchmarks, BRIGHT and BEIR, with 3 different sized Llama models. The authors claim that their methodology performs better than existing state-of-the-art reranking models/methods.

**Strengths:**

- What I appreciated most is the decoupling of analysis and judgement. Breaking down a complex task as so is an elevated approach, especially given the query-based document analysis.
- Inclusion of prompt templates.
- Experimentation across model sizes.

**Weaknesses:**

- The paper could touch base further on the limitations of context length, this is a very intuitive limitation than can be mitigated with a parsing pipeline that creates overlapping shingles for example that are evaluated as separate documents…etc. Even in the presence of 128k context windows, models’ reasoning capabilities tend to decline with really large windows, making it harder to capture a nuanced understanding of the full context, potentially missing some aspects or attending to portions of the context in a manner biased to position.
- Results could provide more example outputs.
- The pipeline might have a scaling limitation with a large number of documents. One thing that can be added to the pipeline is a global topic prompt per doc, followed by a filtering step per query for relevant documents given the latter (possibly even in batches).

**Questions:**

- Have you concluded anything from your experiments that differentiates between reasoning and knowledge heavy reranking tasks?
- Could you elaborate further on why do you think ensembling results are higher? Could it simply be the effect of randomness (especially given the large context window) of multiple retries and the advantage of a majority vote? Have you performed any experimentation to disprove this? majority vote measures with the same model for example?
- Have you explored COT judgement? (beside decoupled analyses)
- Have you explored ensembling with embedding based ranking methods?

---

> ### Author Response · Authors · 2024-11-23
>
> We thank Reviewer WdBJ for the helpful feedback!
>
> * Discuss limitations of context length
>
> Our model is a pointwise reranker which processes one document at a time, thus not suffering from context length limitations unlike listwise rerankers.
>
> * Explore COT judgement instead of decoupled analyses
>
> In our early experiments, we found that CoT judgment, which asks the LLM to perform all the analyses and make a judgment in one take, performs slightly weaker than our current setting. We opt for decoupled analyses also because it maximizes the KV cache we can leverage, hence maximizing the reranker efficiency. That said, we agree that this is a useful baseline to compare with. We have added the relevant motivations in the Method section (Section 2.1).
>
> * Provide more model outputs
>
> We thank the reviewer for the suggestion. We will include more outputs in the Appendix in the revision in addition to what has been presented in Figure 5.
>
> * Conclusion of model performance on reasoning- vs. knowledge-intensive reranking tasks
>
> As shown in Tables 1 and 2, on the BRIGHT benchmark our model outperforms state-of-the-art rerankers by a large margin, while on the BEIR benchmark our model achieves roughly equal performance with those rerankers. The performance gap indicates that our approach is more suitable to reasoning-intensive tasks. This is expected though, because agentic approach is less likely distracted by shallow keyword matching. We chose BRIGHT also because it can help identify models that solely rely on keyword matching. We have added the relevant discussion in the Main Results section (Section 4.1) in the revision.
>
> * Explore ensembling with embedding-based reranking models
>
> We thank the reviewer for the interesting idea! We have added some discussions on this in the Discussion section (Section 4.2) in the revision to frame it as future work.
>
> * Elaborate on why ensembling results are higher.
>
> As extensively discussed in the first research question of Section 4.2, ensembling works because of majority vote: “Intuitively, such ensembling is equivalent to a verification or a majority voting step.”
>
> * One thing that can be added to the pipeline is a global topic prompt per doc, followed by a filtering step per query for relevant documents given the latter (possibly even in batches).
>
> We do not quite follow this idea. Could you elaborate/clarify further? Thanks!
>
> Overall, we really appreciate the reviewer’s insights, which we leverage to refine our paper! We did our best to address every comment and question that the reviewer wrote, and sincerely hope that the reviewer could consider raising our score after seeing the responses and the revised paper. Thank you!

---

> > ### Comment · Reviewer_WdBJ · 2024-11-26
> >
> > Thank you for taking the time to address my concerns and providing further details on your experiments.
> > A few notes on your clarifications:
> > - I understand that you are not using listwise reranking and so context length is less of a concern, but even in pointwise reranking, a single document may still be too large for context, ranking book chapters against a query for example, so how are you addressing these scenarios?
> > -  As for my comment on reasoning vs. knowledge -intensive tasks, it's still unclear to me that your conclusion is true given the narrow test space of simply BRIGHT vs. BEIR. I think it's alright that you haven't explored this in the paper, but in order to make confident claims, you would need to look at some more controlled and hand-crafted experiments. Again, a simple ranking task over book chapters for context-based eval of MMLU high_school_us_history vs. MMLU high_school_physics for example would give a better signal. This "because agentic approach is less likely distracted by shallow keyword matching" particularly doesn't make sense to me given that the same methodology would be applied to both reasoning or knowledge tasks, both require non-shallow retrieval, only that the reasoning-heavy retrieval requires a more-nuanced ability of knowing what to retrieve, in other words, this is the harder task.
> > - Lastly, my suggestion of a global topic and filter is as a possible mitigation for scale, meaning that filtering documents against a global target before running your full pipeline might be helpful. Please feel free to share your own thoughts on how you handle a large number of documents.
> >
> > Thanks!

---

> > > ### Author Response · Authors · 2024-12-03
> > >
> > > We thank Reviewer WdBJ for taking the time to go through our response and provide further feedback!
> > >
> > > * Solution to very long document that does not fit into context length
> > > We think this is out of the scope of our work because it does not happen for the datasets we consider. That said, for such a scenario we can still adapt our method by chunking the document into K acceptable lengths. Each document will receive a probability score to be averaged out across all chunks.
> > >
> > > * Claim about reasoning vs. knowledge-intensive tasks
> > > We thank the reviewer for the suggestion. We will definitely make our claim more accurate in the revision so that it is less from the angle of reasoning vs. knowledge and more from the dimension of complex vs. shallow retrieval.
> > >
> > > * Global topic and filter
> > > We understand your suggestion now about this pre-filtering step. We think it is quite interesting as it could be more accurate than just BM25 keyword matching but much faster than the full pipeline. We will try that out as future work.

---

### Official Review · Reviewer_XWoZ · 2024-11-04

**Soundness:** 3
**Presentation:** 3
**Contribution:** 2
**Rating:** 6
**Confidence:** 4

**Summary:**

JudgeRank represents an advancement in reasoning-intensive document retrieval, demonstrating that structured reasoning approaches can substantially improve reranking performance. The method's success in both specialized (BRIGHT) and general (BEIR) benchmarks validates its effectiveness and generalizability, while the proposed model ensembling techniques offer promising directions for future improvements.

**Strengths:**

1. JudgeRank performs well in BRIGHT and is compatible with other methods in BEIR.
2. JudgeRank is more suitable for (RAG) tasks that require complex reasoning.
3. The paper is well-written.

**Weaknesses:**

1. The computational cost of multiple LLM inferences may not be justified by the performance gains.
2. Sensitivity to prompt wording.
3. Maybe over-complex for simple retrieval tasks?
4. The inability to perform document analysis in advance may results in high costs for each query.

**Questions:**

1. How does your computational efficiency compare to traditional encoding-based frameworks? Could you provide more details such as throughput rate, inference speed, and memory usage?
2. What are your thoughts on using closed-source LLMs like OpenAI's models? There might be situations where you can't obtain "yes" and "no" logits in the top@k (k=20?) logits.
3. Could you provide insights on why the performance difference between Llama 70B and 405B models isn't significant?
4. Perhaps you could use different types of models (maybe same parameters) to create an ensemble?

---

> ### Author Response · Authors · 2024-11-23
>
> We thank Reviewer XWoZ for the insightful feedback.
>
> * Computational cost of multiple LLM inferences is high
>
> We agree that our best setting, which is an ensemble of three LLMs, leads to high computational cost. However, as shown in Table 1, each of the LLMs alone already outperforms the state-of-the-art reranker RankZephyr by a large margin, and the best-performing single model only differs by around 1 point with the ensemble performance. Hence it is not necessary to use multiple LLMs to achieve high performance. As this is a cost vs. quality tradeoff, practitioners can decide on the most suitable configuration based on how sensitive the downstream tasks are to retrieval performance. We have added a discussion on this trade-off in the Discussion section (Section 4.2) in the revision. We have added a discussion on this trade-off in the Discussion section (Section 4.2) in the revision.
>
> * Computational cost is high because document analysis depends on what the query is
>
> As shown in the right subfigure of Figure 4, even without document analysis where the model is judging directly, our models are able to outperform the baselines by a significant margin, showing that we can afford to cut the computational cost by sacrificing around one point in performance. Similar to the point above, this is a cost vs. quality tradeoff that depends on downstream tasks. We have added a discussion on this trade-off in the Discussion section (Section 4.2) in the revision.
>
> * Proposed approach is too complex for simple retrieval tasks
>
> We agree that this is true for our full (i.e., the best) setting. However, as shown in Table 2, our model, with the simple setting of query analysis (which can be KV-cached in advance) + judgment, is still competitive with state-of-the-art rerankers on simple retrieval tasks in BEIR. We have added a discussion on this in the Main Results section (Section 4.1) in the revision.
>
> * Prompts may be sensitive to wording
>
> We thank the reviewer for bringing this up. This is a nice addition to our analysis section. As noted in the second research question of Section 4.2, “binary judgments are sensitive to wording” while our continuous judgment scores are not. To further show this, we conducted an experiment where we use the quantized version of Llama-3.1-70B-instruct to paraphrase all three prompt templates and  rerank with Llama-3.1-8B on BRIGHT. We found that the model achieves an nDCG@10 of 30.36, which is very close to 30.14 as reported in the paper, showing that our approach is not likely sensitive to prompt variations. We have added the discussion in the analysis section, and will include the paraphrased templates in the Appendix for sanity checks in the revision.
>
> * How to adapt JudgeRank to proprietary models like OpenAI’s?
>
> OpenAI allows users to obtain the probabilities of the 5 most likely tokens to generate, which is enough for JudgeRank to work. Even in extreme cases where obtaining probabilities is prohibited, we still proposed a discrete version in the Methodology section (Section 2.3) and presented its performance on the left of Figure 4. This discrete version only requires the judgment (“Yes” or “No”) in words. We can see that the discrete version still outperforms all baselines. We have added a discussion in the Methodology section (Section 2.3) in the revision.
>
> * Comparison of computational efficiency to traditional encoding-based frameworks?
>
> As shown on the right of Figure 4, our Direct Judge setting already outperforms state-of-the-art baselines such as RankZephyr by a significant margin. This setting does not involve any generation of query or document analysis and only generates one token in the judgment step. Therefore, the latency is practically identical to encoding-based models of the same size. Adding query and document analyses increases latency but also boosts reranking effectiveness. We have added a discussion on this trade-off in the discussion section (Section 4.2) in the revision.
>
> * Insights on why the performances of Llama 70B and 405B models do not differ by much
>
> From the benchmark performance comparisons provided by Meta (https://huggingface.co/meta-llama/Llama-3.1-405B), we observe that there are multiple tasks where 405B is not significantly stronger than 70B, such as CommonSenseQA, TriviaQA-Wiki, and BoolQ. We hypothesize that the BRIGHT benchmark also falls in this plateauing category. We have included this observation in the Main Results section (Section 4.1) in the revision.
>
> * Could ensemble different types of models
>
> Thank you for suggesting this experiment. We have added discussions on this in the Conclusion section (Section 6) in the revision.
>
> Overall, we really appreciate the reviewer’s insights, which we leverage to refine our paper! We did our best to address every comment and question that the reviewer wrote, and sincerely hope that the reviewer could consider raising our score after seeing the responses and the revised paper. Thank you!

---

> ### Comment · Reviewer_XWoZ · 2024-11-25
>
> Thank you for your response.
>
> I have a small suggestion regarding the writing:
>
> In Section 2.3, you describe three versions as Discrete, Continuous, and Hybrid. However, in the subsequent discussion (if I am not mistaken), you use "Binary" to refer to "Discrete," including in the figures. This inconsistency might cause some confusion for readers.

---

> > ### Author Response · Authors · 2024-11-25
> >
> > Thank you for pointing this out.
> >
> > We have updated the paper to only use "binary" instead of a mix of "discrete" and "binary".

---

### Official Review · Reviewer_DY8E · 2024-11-09

**Soundness:** 3
**Presentation:** 4
**Contribution:** 2
**Rating:** 5
**Confidence:** 4

**Summary:**

This paper introduces JudgeRank, a new method for LLM-based pointwise passage reranking. The method diverges from existing methods in that they apply LLMs to do passage-level relevance scoring, breaking this process into three LLM-based steps: 1) query decomposition to understand the core problems needing to be solved 2) query-focused summarization of each document and 3) combining these two together outputs together to assign a relevance score to each document. They test their method on two popular reranking benchmarks, finding their method outperforms vanilla first-stage retrieval, and also two baseline LLM methods fine-tuned for this task.

**Strengths:**

The paper is well-written and very easy to follow. The technique is intuitive, taking retrieved top-k documents, and performing passage reranking through the use of instruction-tuned LLMs (LLama-3.1 model family). They do some interesting analysis, such as exploring the similarities between model predictions at different sizes (8b vs 70b vs 405b), exploring ensembling and finding this can improve accuracy, and also doing component-level analysis of the JudgeRank pipeline (comparing different scoring metrics, and also investigating the benefits of this modular reasoning approach).

**Weaknesses:**

Overall, the novelty of this work is somewhat lacking -- the method is effectively comprised of a few intuitive prompting steps which, while effective, are fairly intuitive and don't introduce any fundamentally new way of tackling more challenging reasoning; the authors note this work was inspired by popular techniques such as Chain-of-Thought and LLM-as-a-Judge, and it seems this is more or less a straightforward application of existing ideas to a particular use case.

I think some additional steps/experiments could help to bolster the paper and add further takeaways; some concrete directions might be (1) to dig deeper on the base model selection, understanding if/how other model families (both open and propietary models, so long as they provide relevant confidence scores) compare on this task, (2) further exploring aggregation/ensembling methods with better cost:performance ratio (e.g., what happens if you ensemble a mixture of several smaller models from different families), and (3) investigating more 'adaptive' prompting configurations targeted to each use case (e.g. you note: "Leveraging LLMs in a zero-shot setting allows us to flexibly define the relation between the query and the document....this flexibility is important because the user may define either “document that supports a query” or “document that refutes a query...”).

It would also be interesting to understand what the current 'ceiling' is for this task. It seems there is a first-stage BM25 step applied, and I'm curious what impact this has on final performance -- ie, what are the implications if you vary k during the first-stage (from latency/scaling to performance); could your method potentially perform better without the first-stage retrieval step?

**Questions:**

**Base models:** have you explored base models other than Llama3.1?

**Prompt design and variants:** How much impact do the prompts have on JudgeRank’s performance? The paper mentions potential prompt variations -- when would different prompts be applicable, and how might they influence retrieval effectiveness?

**Ensembling** Maybe I missed this, but can you share the model ensemble performance on the BEIR benchmark?

**Listwise reranking:** I understand this is somewhat orthogonal to your approach, but do you have any thoughts on how you might adapt your technique + intuition to listwise reranking (instead of pointwise).

**Typos / Grammar / Presentation**
- l138 - mimicking
- l312 - performing
- l271 - best-first retrieval? best retrieval-first model?
- Table 2 - I might suggest presenting the models in the same order as the prior table. Also, it might be helpful if you explicitly re-state (maybe in the caption?) the details of this JudgeRank variant (that it's the 70B variant w/ ensemble)

---

> ### Author Response · Authors · 2024-11-23
>
> We thank Reviewer DY8E for the insightful feedback!
>
> * The approach lacks novelty because it only adapts CoT and LLM-as-a-Judge to reranking
>
> We thank the reviewer for interpreting our approach from this angle. Our novelty lies in its simplicity and effectiveness on reasoning-intensive tasks. More specifically, our work is the first to show that a zero-shot pointwise reranker can outperform a well-trained listwise reranker by a very significant margin. This is important because it challenges the common belief that listwise rerankers in general perform better than pointwise rerankers. In fact, for the reasoning-intensive BRIGHT benchmark, listwise reranker (e.g., RankZephyr) suffers from context length limit, domain shift from training data, and a lack of reasoning steps. We have added the relevant discussion in the Introduction section (Section 1) to further highlight the novelty in the revision.
>
> * Explore LLMs from other model families and ensemble them
>
> Thank you for suggesting this ablation study. We have added some discussions on this in the Conclusion section (Section 6) in the revision to frame it as future work.
>
> * Investigate more adaptive prompting configurations
>
> We totally agree that if we were to further adapt the templates to each specific task, it might lead to better performance. However, we do note that there is a trade-off between performance and generalization because such adaptation also makes prompt transfer harder to future tasks. As shown in Figure 2 and discussed in Section 2.2, our prompt templates only contain placeholders like {query name}, {doc name} and {definition of relevance} because such setting maximizes the generalizability of this approach. We have added a discussion of this trade-off in the footnote of Section 2.2 to make practitioners aware that each prompt can be further adapted to specific tasks in the revision.
>
> * How does prompt variations influence reranking performance
>
> We thank the reviewer for bringing this up and think this is a nice addition to our analysis section. As noted in the second research question of Section 4.2, “binary judgments are sensitive to wording” while our continuous judgment scores are not. To further show this, we have conducted a new experiment where we use the quantized version of Llama-3.1-70B-instruct to paraphrase all three prompt templates, namely query analysis, document analysis, and judgment, and conducted the reranking experiment with Llama-3.1-8B. We found that the model achieves an nDCG@10 of 30.36, which is very close to 30.14 as reported in the paper, showing that our approach is not likely sensitive to prompt variations. We have added the discussion in the analysis section, and included the paraphrased templates in the Appendix for sanity checks in the revision.
>
> * Model ensemble on the BEIR benchmark
>
> We did not perform model ensemble on BEIR because of its prohibitive computational cost. There are 5 tasks that have over 5K queries, among which 2 of them have over 10K. Despite the cost, we agree that given enough time it is a good idea to include in the final version of the paper.
>
> * Intuition on how to adapt JudgeRank to listwise reranking
>
> Thank you for the interesting idea! One could generate an analysis for each document separately, and then use an concatenation of each document and its analysis as input to the LLM with the listwise prompt. In other words, the input to the listwise reranker would be {listwise_prompt, query, query_analysis, doc_1, doc_1_analysis, ..., doc_k, doc_k_analysis}. One drawback for this approach is that listwise rerankers are already struggling with long context length — including all the analyses only exacerbates this situation.
>
> * Typos / Grammar / Presentation
>
> Thank you for the suggestions! We have corrected these in the revision.
>
> Overall, we really appreciate the reviewer’s insights, which we leverage to refine our paper! We did our best to address every comment and question that the reviewer wrote, and sincerely hope that the reviewer could consider raising our score after seeing the responses and the revised paper. Thank you!

---

> ### Author Response · Authors · 2024-12-03
>
> Dear Reviewer DY8E,
>
> As a friendly reminder, today, December 2, marks the final day for reviewers to communicate any messages to the authors. We sincerely hope that we have adequately addressed all your questions and concerns.
>
> If you feel that your feedback has been satisfactorily met, we would appreciate it if you could consider updating your score. Thank you for your time and contributions!
>
> Best Regards,
> JudgeRank Authors

---

### Author Response · Authors · 2024-11-23

We thank the reviewers for their constructive feedback!
We appreciate that overall the reviewers think that:
1. The paper is well-written;
2. The proposed approach performs well on the BRIGHT benchmark and is suitable for reasoning-intensive retrieval tasks;
3. The approach is simple, intuitive, effective, and able to generalize well across diverse reranking tasks;
4. The work presents interesting ablation studies;
5. The decoupling of analyses and judgment is considered a significant contribution.

We posted a response for each review and uploaded a revised paper to accommodate reviewers' suggestions.

---

### Meta-Review · Area_Chair_LLwd · 2024-12-20

**Metareview:**

This paper introduces JudgeRank, a novel LLM-based method for pointwise passage reranking. It employs a three-step process: query decomposition to identify core issues, query-focused document summarization, and combining these outputs for relevance scoring. Tested on BRIGHT and BEIR benchmarks, JudgeRank outperforms vanilla retrieval and fine-tuned LLM baselines.

Strength:
- The paper is very well-written and easy to follow.
- The authors presents an intuitive technique leveraging instruction-tuned LLMs for passage reranking of top-k retrieved documents.

Weakness:
- The exploration of other model families and ensembling is framed as future work in the revised conclusion. While this acknowledges the limitation, the lack of immediate experimental results diminishes the effectiveness and reliability of the proposed approach.

- Reviewer 27LA raised the score from 3 to 5 but highlighted that several concerns remain unaddressed:

 1. The absence of results for removing the query or document analysis step individually.
2. A lack of evaluation for combining document analysis and judgment into a single prompt.
 3. No detailed analysis of the impact of "{definition of relevance}" on reranking performance.

**Additional Comments On Reviewer Discussion:**

Reviewer DY8E's concerns still seem unresolved (despite no reply from the reviewer), leading to a rating below the acceptance threshold.

- Reviewer DY8E suggested exploring other model families and ensembling. The authors addressed this by framing it as future work in the revised conclusion without provide immediate experimental results, this limits the effectiveness and reliability of the proposed approach.

Additionally, Reviewer 27LA raised the score from 3 to 5 but pointed out that some of the concerns remain unaddressed. These include:
- The results of removing the query or document analysis step separately.
- Combining document analysis and judgment into a single prompt.
- The impact of "{definition of relevance}" on reranking performance.

---

### Decision · Program_Chairs · 2025-01-22

Reject